# Preventing the Next Pandemic: Is Live Vaccine Efficacious against Monkeypox, or Is There a Need for Killed Virus and mRNA Vaccines?

**DOI:** 10.3390/vaccines10091419

**Published:** 2022-08-29

**Authors:** Abdelaziz Abdelaal, Abdullah Reda, Basant Ismail Lashin, Basant E. Katamesh, Aml M. Brakat, Balqees Mahmoud AL-Manaseer, Sayanika Kaur, Ankush Asija, Nimesh K. Patel, Soney Basnyat, Ali A. Rabaan, Saad Alhumaid, Hawra Albayat, Mohammed Aljeldah, Basim R. Al Shammari, Amal H. Al-Najjar, Ahmed K. Al-Jassem, Sultan T. AlShurbaji, Fatimah S. Alshahrani, Ahlam Alynbiawi, Zainab H. Alfaraj, Duaa H. Alfaraj, Ahmed H. Aldawood, Yub Raj Sedhai, Victoria Mumbo, Alfonso J. Rodriguez-Morales, Ranjit Sah

**Affiliations:** 1Postgraduate Medical Education, Harvard Medical School, Boston, MA 02115, USA; 2School of Medicine, Boston University, Boston, MA 02118, USA; 3Tanta Research Team, Tanta 31527, Egypt; 4Faculty of Medicine, Tanta University, Tanta 31527, Egypt; 5Faculty of Medicine, Al-Azhar University, Cairo 11884, Egypt; 6Faculty of Medicine, Banha University, Benha 13511, Egypt; 7Faculty of Medicine, Zagazig University, Ash Sharqia Governorate, Zagazig 44519, Egypt; 8Jordan University Hospital, Amman 11942, Jordan; 9School of Medicine, University of Jordan, Amman 11733, Jordan; 10Department of Internal Medicine, West Virginia University, Morgantown, WV 26506, USA; 11Department of Internal Medicine, Virginia Commonwealth University, Richmond, VA 23284, USA; 12Molecular Diagnostic Laboratory, Johns Hopkins Aramco Healthcare, Dhahran 31311, Saudi Arabia; 13College of Medicine, Alfaisal University, Riyadh 11533, Saudi Arabia; 14Department of Public Health and Nutrition, The University of Haripur, Haripur 22610, Pakistan; 15Administration of Pharmaceutical Care, Al-Ahsa Health Cluster, Ministry of Health, Al-Ahsa 31982, Saudi Arabia; 16Infectious Disease Department, King Saud Medical City, Riyadh 11564, Saudi Arabia; 17Department of Clinical Laboratory Sciences, College of Applied Medical Sciences, University of Hafr Al Batin, Hafr Al Batin 39831, Saudi Arabia; 18Drug & Poison Information Center, Pharmacy Department, Security Forces Hospital Program, Riyadh 11564, Saudi Arabia; 19Outpatient Pharmacy, Dr. Sulaiman Alhabib Medical Group, Diplomatic Quarter, Riyadh 91877, Saudi Arabia; 20Department of Internal Medicine, College of Medicine, King Saud University, Riyadh 11362, Saudi Arabia; 21Division of Infectious Diseases, Department of Internal Medicine, College of Medicine, King Saud University, Riyadh 11451, Saudi Arabia; 22Infectious Diseases Section, Medical Specialties Department, King Fahad Medical City, Riyadh 12231, Saudi Arabia; 23Department of Nursing, Maternity and Children Hospital, Dammam 31176, Saudi Arabia; 24Molecular Diagnostic Laboratory, Dammam Regional Laboratory and Blood Bank, Dammam 31411, Saudi Arabia; 25Division of Pulmonary Diseases and Critical Care Medicine, University of Kentucky, Bowling Green, KY 40292, USA; 26Coast General Teaching and Referral Hospital, Mombasa P.O. Box 90231-80100, Kenya; 27Latin American Network on Monkeypox Virus Research (LAMOVI), Pereira 66001, Colombia; 28Institución Universitaria Visión de las Américas, Pereira 12998, Colombia; 29Grupo de Investigación Biomedicina, Faculty of Medicine, Fundación Universitaria Autónomade las Américas, Pereira 66003, Colombia; 30Master of Clinical Epidemiology and Biostatistics, Universidad Científica del Sur, Lima 4861, Peru; 31Tribhuvan University Teaching Hospital, Institute of Medicine, Kathmandu 44600, Nepal

**Keywords:** monkeypox, vaccine, outbreak

## Abstract

(1) Background: The monkeypox virus (MPV) is a double-stranded DNA virus belonging to the Poxviridae family, Chordopoxvirinae subfamily, and Orthopoxvirus genus. It was called monkeypox because it was first discovered in monkeys, in a Danish laboratory, in 1958. However, the actual reservoir for MPV is still unknown. (2) Methods and Results: We have reviewed the existing literature on the options for Monkeypox virus. There are three available vaccines for orthopoxviruses—ACAM2000, JYNNEOS, and LC16—with the first being a replicating vaccine and the latter being non- or minimally replicating. (3) Conclusions: Smallpox vaccinations previously provided coincidental immunity to MPV. ACAM2000 (a live-attenuated replicating vaccine) and JYNNEOS (a live-attenuated, nonreplicating vaccine) are two US FDA-approved vaccines that can prevent monkeypox. However, ACAM2000 may cause serious side effects, including cardiac problems, whereas JYNNEOS is associated with fewer complications. The recent outbreaks across the globe have once again highlighted the need for constant monitoring and the development of novel prophylactic and therapeutic modalities. Based on available data, there is still a need to develop an effective and safe new generation of vaccines specific for monkeypox that are killed or developed into a mRNA vaccine before monkeypox is declared a pandemic.

## 1. Monkeypox: The Past and Now

The monkeypox virus (MPV) is a double-stranded DNA virus belonging to the Poxviridae family, Chordopoxvirinae subfamily, and Orthopoxvirus genus [1]. It was called monkeypox because it was first discovered in monkeys, in a Danish laboratory, in 1958 [2]. However, the actual reservoir for MPV is still unknown [3]. On the other hand, the first reported case in humans was in an infant in the Democratic Republic of the Congo (DRC) in 1970 [4]. Then, it spread among other African countries, mainly Central and West Africa. Finally, in 2003, it started to spread outside Africa [5], reaching the United States by importing animals from Ghana [6]. Genetically, MPV has two clades: the West African and the Central African (Congo Basin) clade [7].

A systematic review showed that the number of monkeypox cases was highest in DRC, particularly in Central Africa, with 38, 520, and 85 confirmed cases reported during 1970–1979, 1990–1999, and 2010–2019, respectively. Outside Africa, mainly in the United States, MPV was detected in 47 cases from 2000–2009 [6]. That being said, MPV started re-emerging in 2022, affecting numerous patients with no connection worldwide. As of 3 July 2022, MPV was confirmed in 5783 cases in more than 52 nonendemic countries, with the highest numbers in the United Kingdom, Germany, Spain, France, the USA, and Portugal, in decreasing order [8].

This virus infects humans, causing monkeypox, which is quite similar to the disease caused by the smallpox virus. MPX has similar presentation and severity to smallpox but is associated with lower mortality (3–6% in the recent multicountry outbreak) and less human-to-human transmission [9,10]. Moreover, smallpox vaccines have been shown to provide cross-protection to MPV in 85% of cases [11]. As a result, the incidence rate of the MPV is highest among unvaccinated patients [6].

Current vaccines, although they provide cross-protection against monkeypox, are not specific for the causative virus and their efficacy in the light of the recent multicountry outbreak is still to be confirmed. In addition, as a consequence of the eradication and cessation of smallpox vaccination for four decades, MPV found an opportunity to re-emerge, but with different characteristics.

## 2. Prevention and Prophylaxis

Our immune system includes innate and adaptive components; innate immunity is the first defence line against infections, represented by physical, chemical, and microbiological barriers. Meanwhile, adaptive immunity comes after a previous infection to act specifically toward the previous immunogenic antigen [12]. In this regard, vaccines protect against such antigens without prior exposure, especially in prevalent infectious diseases.

One of the recent examples is the COVID-19 pandemic, which did not only affect the preparedness of the global healthcare system, but also affected it at the individual level, limiting the daily life activities of many people. The burden of COVID-19 pandemic was not reduced, in terms of associated morbidity and mortality, until effective vaccines were developed and disseminated. The herd immunity threshold level is calculated mathematically for each population; if this level is achieved, it will lead to herd immunity or population immunity, which protects unvaccinated individuals of the people indirectly when a certain percentage of the same population has been vaccinated and infected [13].

Herd immunity is a remarkable measure of vaccine efficacy. It helps to control and contain infectious diseases. However, the unknown mechanism of herd immunity led the researchers to understand its dynamics better to use it wisely in this era of newly emerging and re-emerging diseases [14], especially if the vaccines cannot cover the whole population for many reasons, including economic, religious, or medical reasons.

## 3. What Are the Different Types of Vaccines?

Live (-attenuated) vaccines are produced from weakened live viruses; they replicate in the host and create a milder form of the natural infection resulting in an immune response identical to natural infection. They do, however, have a slight risk of fatal infections when they replicate uncontrollably and, therefore, cannot be given to immunocompromised patients. There is also a risk of vertical and horizontal spread caused by contact with contaminated material [15]. The differences between live-attenuated and killed vaccines are highlighted in Table 1.

Inactivated virus vaccines consist of whole viruses that have been treated with chemicals or heat. Chemicals, such as phenol and formaldehyde, and heat cause the denaturation of the surface proteins, inactivating them and making them noninfective. The treatment of the whole virus leaves some unchanged epitopes, which maintain some of their integrity and evoke an adaptive immune response by initiating antibody formation [16]. When injected, phagocytic immature dendritic cells divide the virus into smaller antigenic fragments, which are presented as antigenic fragments on the surface of MHC cells leading to the activation of B cells and T helper cells. Inactivated vaccines are safe with no adverse reactions. They are also noninfectious, heat-stable, and nontransmissible of the disease, as they cannot replicate. However, they are insufficient as a single dose and they usually do not provide immunity that is as strong as live vaccines; therefore, booster doses may be needed periodically to produce a sufficient immune response [17].

There are two types of mRNA vaccines: self-amplifying and nonreplicating. Self-amplifying vaccines contain codes for both the target antigen and the replicase complex, enabling the vaccine’s amplification. Nonreplicating RNA have codes for the target antigen alone and untranslated regions. Once inside the cell, the mRNA replicates to form a protein triggering an immune reaction and enabling the stimulation of both the innate and adaptive immune systems [18]. mRNA vaccines have many advantages; they can effectively activate both humoral and cell-mediated immunity, producing durable immune memory. Production of these vaccines is fast and inexpensive; thus, a large number can be made in a short time. In addition, they are stable, carry no risk of infection, do not integrate with the host genome, and are naturally degraded [19]. Noteworthily, a concern associated with this type of vaccine is the unknown risk for long-term adverse events [20].

## 4. Available Vaccines for Monkeypox Virus

Preventing the transmission and infection of MPV is associated with different challenges. The primary strategy of prevention would be vaccination. However, no specific vaccines have been developed for MPV. On the other hand, many previous studies conducted in the aftermath of smallpox infections demonstrated that smallpox vaccines could be effectively used to protect against the MPV [21]. For instance, older evidence indicates that smallpox vaccines might induce a cross-reaction with the MPV and can be efficacious as 85% in preventing the infection [22]. However, it should be noted that not all of these vaccines are primarily used because of their associated side effects and complications [23], which will be discussed in the following section.

Evidence indicates that remarkable advances have been introduced to vaccine technology after smallpox to enhance the safety and efficacy of these vaccines. Three generations of smallpox vaccines were developed accordingly with enhanced features [24]. For instance, first-generation smallpox vaccines were usually propagated through calfskin and collected at calf lymph. However, using these vaccines from the Smallpox Eradication Program (SEP) kept in national and WHO reserves is not currently recommended for MPV interventions. On the other hand, second-generation smallpox vaccines are produced with modern, sound manufacturing practices and propagated in tissue cell cultures. Accordingly, these vaccines have a reduced risk of contamination by surrounding factors.

Moreover, it has been shown that both of these generations are usually associated with an increased risk of developing adverse events as they have a replication-competent vaccinia virus [25,26]. Third-generation smallpox vaccines are manufactured similarly to second-generation ones. However, they have enhanced safety profiles because they contain attenuated vaccinia viruses rather than replication-competent vaccinia ones [5,27,28]. Table 2 summarises the structure and efficacy of currently available vaccines to help prevent MPV infection.

The Food and Drug Administration approved the second-generation smallpox vaccine ACAM2000 to be used in emergencies and outbreaks of smallpox for postexposure prophylaxis. Accordingly, it has been purchased for the Strategic National Stockpile (SNS) and was used for different population groups [1]. Furthermore, in 2019, MVA-BN or JYNNEOS^TM^ was approved in the United States and Canada after various animal studies, and clinical trials showed the high efficacy and safety of this vaccine, which can also be used in different population groups for the prevention of MPV infection [2,29,30,31,32,33,34,35,36]. Furthermore, JYNNEOS^TM^ was approved based on the data suggesting its noninferior immunogenicity to other smallpox vaccines, its efficacy in animals, and its safety profile in humans [27,37,38,39,40,41]. Moreover, LC16 has been approved in the United States by the emergency investigational new drug program of the US Food and Drug Administration and in Japan, secondary to the reported protective effects in animal models and immunogenicity in human studies [41,42,43]. Although LC16 is the only approved smallpox vaccine for use in children, there are no data on its efficacy in preventing MPV infection.

It should be noted that these vaccines are usually efficacious in preventing MPV infection when used as pre-exposure approaches. Nevertheless, experts demonstrated that postexposure vaccination could also intervene against the development of severe diseases or reduce the severity of the conditions of infected MPV patients [44]. In this context, it can be suggested that it is better to be vaccinated sooner after exposure. Evidence from the Centers for Disease Control and Prevention (CDC) recommends that vaccinating exposed individuals should be conducted within four days of exposure to prevent the onset of the disease [45]. Therefore, the beginning of the illness may be inevitable in individuals that did not receive the vaccination within this period. However, vaccination within the first two weeks might intervene against developing the severe disease [46].

There is also evidence that the protective efficacy of smallpox vaccines usually fades over time. However, it has been shown that postvaccination protection might last for up to 20 years. Besides, although the protective efficacy of the smallpox vaccine fades with time, vaccinated individuals will still have lifelong protection against developing a severe disease secondary to the presence of memory B and T cells [41,47]. Therefore, immunity against the MPV should be expected in previously vaccinated individuals against smallpox [48]. The CDC also recommends that vaccination be given to individuals exposed to the MPV and not vaccinated within the last three years [45]. In this context, it has been demonstrated that receiving the smallpox vaccine as soon as possible would be more effective in preventing MPV infection [49].

## 5. Efficacy of Available Vaccines

There are three available vaccines for orthopoxviruses—ACAM2000, JYNNEOS, and LC16—with the first being a replicating vaccine and the latter being non- or minimally replicating.

In 2015, ACAM2000 was approved by the FDA and licensed in the USA for smallpox and monkeypox and was the only available monkeypox vaccine in the USA from 2015–2019 [24,50]. ACAM2000 was developed using cell culture techniques in France and the USA. It was approved for people aged 18–64 years old. It is a second-generation, live-attenuated, plaque-purified, replication-competent vaccinia virus vaccine produced by Emergent BioSolutions. The vaccine is administered percutaneously using a bifurcated needle using the scarification technique by multiple inoculations of the preparation into the skin surface. It is a single-dose vaccine with peaked protection produced within 28 days of immunisation. Booster doses are given every three years for people exposed to high-virulent strains of orthopoxviruses and ten years for those exposed to low-virulent strains (i.e., *Vaccinia virus* or *Cowpox virus*) [50].

In 2013, the MVA-BN vaccine was authorised in Canada to be used for protection against smallpox; later, in 2019, it was approved for monkeypox use in Canada and licensed by the FDA for smallpox and monkeypox prevention in high-risk people aged 18 and older in the USA. The MVA-BN vaccine is a third-generation, live-attenuated, nonreplicating Ankara vaccine produced by Bavarian Nordic [51]. It is a two-dose vaccine given 28 days apart, providing protection two weeks after the second dose. Some clinical trials reported a strong antibody response after administering the first dose [38]. Immunity is produced two weeks after the administration of the second dose. A booster dose is required every two years for people with high-virulent orthopoxviruses and every ten years for those in contact with low-virulent strains [52].

The LC16 vaccine is a third-generation vaccine produced by KM Biologics. It was developed and licensed for smallpox use in Japan (1975) and later licensed in the USA in 2014; however, it has not yet received a license for monkeypox use [24]. It is a live-attenuated minimally replicating vaccine developed using cell culture techniques [42]. LC16 is produced from the Lister strain containing a deleted mutation in its immunogenic membrane protein B5R. The multidose vaccine is given percutaneously to people of all ages, including infants and children, using a bifurcated needle [24].

It is essential to consider the reactogenicity, safety, and vaccine-associated adverse events to establish the best effectiveness and determine the best vaccine, especially for high-risk and vulnerable groups. For instance, different types of vaccines, including nonreplicating (such as JYNNEOS^TM^), minimally replicating (such as LC16), and replicating vaccina-based vaccines (such as ACAM2000), can be considered for healthy adults [41]. On the other hand, adults with contraindications to receiving ACAM2000 and other replicating vaccines (due to atopic dermatitis, immunosuppression therapies, immune deficiencies, or hematopoietic stem cell transplantation recipients) should receive other nonreplicating specific MPV vaccines, such as JYNNEOS^TM^ [28,32,37,47]. Furthermore, pregnant and breastfeeding women should be indicated to receive minimally replicating (LC16) or nonreplicating MPV vaccines (JYNNEOS^TM^) regarding pre- and postexposure vaccination [41]. Similarly, the options mentioned above are preferred in postexposure prophylaxis of children. However, it should be noted that JYNNEOS^TM^ is only authorised for adults >18 years old. Accordingly, its administration in children is considered off-label. Finally, ACAM2000 is not recommended for preventing MPV infection in children due to unavailable data [53]. That indicates that further studies providing enhanced evidence in this context are encouraged.

## 6. Reported Vaccine-Associated Complications

A thorough comparison between available vaccines in terms of safety and reported complications is provided in Table 3.

### 6.1. ACAM2000 (Live-Attenuated, Replicating Vaccine)

ACAM2000 is contraindicated in persons with conditions associated with immunosuppression (e.g., leukaemia, lymphoma, human immunodeficiency virus (HIV) infection, and acquired immune deficiency syndrome (AIDS)) [50]. Patients with metastatic malignancy, transplant recipients, autoimmune disease (with immunodeficiency as a clinical component), and those undergoing therapy that suppresses the immune system such as corticosteroids (≥2 mg/kg body weight for ≥2 weeks), stem cell transplant recipients <24 months, alkylating agents, TNF inhibitors, antimetabolites, and radiation are not advised to receive the vaccine [1].

Individuals who have serious skin conditions such as allergy to any component of ACAM2000, eczema, dermatitis, psoriasis, or other active exfoliative skin conditions (for instance, varicella-zoster virus infection, herpes simplex virus infection, burns, impetigo, severe acne, severe diaper rash with extensive areas of peeled skin) are at increased risk of complications and, if needed, should be vaccinated with caution [1,24]. Further, it should not be used in pregnant women and infants below <1 year. In addition, it should be avoided in breastfeeding and young children. ACIP does not recommend nonemergency vaccination with ACAM2000 for adolescents aged <18 years [1].

Data from epidemiologic studies and clinical trials indicate that vaccination may increase the risk for myopericarditis. Although the associated risk factors for myopericarditis after smallpox vaccination have not been identified, the impacts of myopericarditis are still more severe in individuals with known heart disease and cardiac risk factors than in individuals without these conditions [46,54,55].

Heart diseases that are mainly associated with high risk are coronary artery disease (or cardiomyopathy with or without symptoms), diabetes, hypertension, smoking, hypercholesterolemia, or heart disease at the age of 50 years in a first-degree relative [1,46,54,55]. Neurological symptoms such as postvaccinal encephalitis, encephalomyelitis, or encephalopathy have been reported [1,46,54].

ACAM2000 common adverse events include injection site reactions, lymphadenitis, and constitutional manifestation, such as fever, fatigue, headache, malaise, and myalgia. Severe adverse events related to ACAM2000 include relatively uncommon, generalised reactions such as progressive and generalised vaccinia, skin infections, erythema multiforme including eczema vaccinatum, and Stevens–Johnson syndrome [1,46]. In addition, an accidental infection can occur, most commonly through inoculation of the eyelids or conjunctiva. However, unintentional infection of other body sites such as the mouth, lips, genitalia, and anus is also possible. That occurred in the majority of patients 5–12 days after vaccination. If vaccination is being considered for someone who lives in the same household or has close contact with a vulnerable person, ACAM2000 should be avoided if possible; otherwise, the vaccinee must take caution to avoid contact with newborns, children, pregnant women, or other members of the household [1,54].

Direct contact between a mother and her infant could result in the transmission of the live vaccinia virus. It should be acknowledged that severe health problems can occur in unvaccinated people who are accidentally infected by someone who has recently received the vaccine. Unvaccinated individuals (i.e., pregnant women, people with heart or immune diseases, or individuals with skin problems such as eczema, dermatitis, or psoriasis) are at an increased risk of serious problems if they become infected with MPV either through vaccination or through close contact with a vaccine recipient [1,54].

### 6.2. MVA-BN (Live-Attenuated, Nonreplicating Vaccine)

Individuals who have experienced allergic reactions to any MVA-BN vaccine components should use it cautiously. MVA-BN has been studied in people with atopic dermatitis and has shown immunogenicity in eliciting a neutralising antibody response without causing any significant safety concerns [50]. However, atopic dermatitis patients may experience more severe local skin reactions (such as redness, swelling, and itching) as well as other general manifestations (such as headache, muscle pain, and feeling sick or tired), as well as a flare-up or exacerbation of their skin condition [24,46].

Healthcare professionals and vaccine administrators should be prepared to manage anaphylactic reactions following the administration of MVA-BN. Injection site reactions were the most common adverse effects (>1 in 10 vaccinees). The local manifestations included pain, redness, swelling, induration, and itching; systemic manifestations included (muscle pain, headache, fatigue, nausea, myalgia, and chills) [24]. Immunocompromised individuals, including those on immunosuppressive therapy, may have a reduced immune response to MVA-BN due to their deficient immune system defences. Clinical studies have not detected any risk of myopericarditis in recipients of MVA-BN [50,55].

There is insufficient human data on pregnant women’s administration to determine vaccine-associated pregnancy risks. However, animal models such as rats and rabbits have shown no evidence of prenatal harm. MVA-BN safety and efficacy in breastfeeding women have not yet been studied. MVA-BN is not known to be excreted in human milk, and there are no data to measure the effects of MVA-BN on milk production or the safety of MVA-BN in breastfed infants. The MVA-BN vaccine is replication-deficient; thus, it should not present a risk of transmission to breastfed infants. However, caution should always be attempted when considering the administration of MVA-BN to breastfeeding women [50,54].

### 6.3. LC16 Vaccination (Freeze-Dried, Live-Attenuated Vaccine)

The LC16 vaccine should be used with caution in individuals who are immunocompromised, have atopic dermatitis during pregnancy, or have an allergic reaction to any vaccine component. Healthcare professionals and vaccine administrators must be prepared to treat anaphylactic reactions after administering LC16 [24,46,54].

Minor side effects of the LC1618 vaccine include lymphadenopathy, fatigue, fever, rash, erythema, and swelling at the injection site. Side effects are significantly more common in primary vaccinated than in re-vaccinated individuals. However, there have been no serious adverse events reported [24,54].

## 7. Is There a Need for Another Vaccine?

Monkeypox cases are increasing worldwide, raising concerns that the virus, such as SARS-CoV-2, could become a pandemic with disastrous consequences on the already exhausted healthcare system. Licensed smallpox vaccines provide MPV cross-protection, including ACAM2000 and JYNNEOS [56,57].

The Vaccine Adverse Event Reporting System (VAERS) obtained 1149 reports regarding ACAM2000 administration, 169 (14.7%) of which were severe (leading to permanent disability, hospitalisation or prolongation of hospitalisation, life-threatening illness, or death), including one death. Cardiovascular and infectious conditions were the two most common categories for serious reports [58].

From 2009 to 2017, electronic records surveillance of the entire vaccinated military service population revealed rates of myopericarditis, defined neurological events, and overall cardiac events consistent with prior passive surveillance studies using Dryvax or ACAM2000 smallpox vaccines. In addition, myopericarditis increased 50-fold following the replacement of the Dryvax smallpox vaccine with the ACAM2000 smallpox vaccine as a mandatory vaccination for deployed military personnel [59].

In September 2019, the United States Food and Drug Administration (FDA) approved JYNNEOS to prevent monkeypox and smallpox in adults. The approval for monkeypox indication was based on survival data from nonhuman primary research. The survival rate in animals vaccinated with JYNNEOS ranged from 80% to 100%, compared with 0% to 40% in the control group. In the plaque reduction neutralisation test, JYNNEOS revealed noninferiority in immunogenicity. In November 2021, the Advisory Committee on immunisation practices (ACIP) approved JYNNEOS as an alternative to ACAM2000 for initial vaccination and booster doses (Table 4). The most common adverse events of JYNNEOS during the trials were injection site reactions, headache, nausea, throat tightness, myalgia, and chills [60]. The presence of a “take” (i.e., the formation of intradermal scarification, resulting in a significant cutaneous reaction “pustule” at the vaccination site) was used as a marker for vaccine efficacy and taken as evidence of protection against smallpox. Due to a lack of replication in human cells, JYNNEOS does not induce severe cutaneous responses [54].

Based on the observations mentioned above, the efficacy of available vaccines is still questioned. It is thought not to provide the expected and required efficacy to limit the spread of the disease, besides the frequent occurrence of vaccine-associated severe complications that could surpass MPV in terms of associated morbidity and mortality. Therefore, we are in dire need of developing new vaccines that are safer, more efficacious, and highly specific for the MPV. In this regard, we recommend developing and testing newer killed and/or mRNA vaccines to overcome the downsides of available ones before monkeypox is declared a pandemic and preparedness activities become more challenging to implement. Nonetheless, until such vaccines are tested and made available, the MVA-BN vaccine can be used to limit the spread of the disease because of its efficacious and safer profile compared with the ACAM2000 vaccine.

## 8. Conclusions

Monkeypox is a vesiculopustular rash illness transmitted to humans through direct contact with an infected person or animal or through contact with virus-contaminated material, with a case fatality rate of 3–6%. The decline in herd immunity caused by the cessation of smallpox vaccination has created a landscape for MPV resurgence between humans and potential MPV animal reservoirs, providing an immunological and ecological niche for MPV to re-emerge. In addition, it is no longer a “rare”, viral, zoonotic disease that primarily affects remote areas of Central and West Africa.

Concerns have recently been raised about the emergence of the human MPV. A multicountry outbreak of the MPV has gained global attention. More intensive surveillance and study on MPV epidemiology, ecology, and biology in endemic locations are required to understand and control its rapid human-to-human transmission.

Smallpox vaccinations previously provided coincidental immunity to MPV. ACAM2000 (a live-attenuated replicating vaccine) and JYNNEOS (a live-attenuated, nonreplicating vaccine) are two US FDA-approved vaccines that can prevent monkeypox. However, ACAM2000 may cause serious side effects, including cardiac problems, whereas JYNNEOS is associated with fewer complications. The recent outbreaks across the globe have once again highlighted the need for constant monitoring and the development of novel prophylactic and therapeutic modalities. Based on available data, there is still a need to develop an effective and safe new generation of vaccines specific for monkeypox that are killed or developed into mRNA vaccines before monkeypox is declared a pandemic.

## Figures and Tables

**Table 1 vaccines-10-01419-t001:** The differences between live-attenuated and killed vaccines.

	Live	Killed
Virus	Weakened Live virus	Entire virus is killed
Replication	Can replicate and mimic natural infection	Do not replicate
Immunity	Greater and longer duration	Lower and Shorter duration
Immune response	Cell-mediated	Humoral
Adjuvant	Not needed	Needed
Ig produced	IgA and IgG	IgG
Virulence	May reverse	No virulence
Booster	Not required	Required
Spread of strain	Vertical and horizontal spread	Not possible

**Table 2 vaccines-10-01419-t002:** Structures, generations, and effectiveness of reported smallpox vaccines for preventing monkeypox infections.

Vaccine	Generation	Effective for	Use for Monkeypox	Structure	Injection Materials	Presentations
LC16	3rd generation	Infants, children, and adults (all ages)	No	Minimally replicating vaccinia virus	Bifurcated needle	Freeze-dried Multidose vials
ACAM2000	2nd generation	Adults (18–64)	USA: postexposure prophylaxis	Propagated in tissue cell culture and produced under good manufacturing practices (live, replication-competent virus)	Bifurcated needle	Freeze-dried Multidose vials
JYNNEOSTM/MVA-BN	3rd generation	General adult population	Yes (USA, UK, Canada)	Nonreplicating vaccinia virus	Needle and syringe (subcutaneous administration)	Liquid frozen or lyophilised (freeze-dried) Single-dose vials (Multidose vials possible)
Vaccinia (Dryvax, Lister, Copenhagen)	1st generation	-	No	Several different strains of vaccinia virus propagated in calf lymph (live, replication-competent virus)	Bifurcated needle	Liquid frozen or lyophilised vials or ampoules

**Table 3 vaccines-10-01419-t003:** Contraindication to the administration of ACAM2000, MVA-BN, and LC16 vaccines.

Contraindications	ACAM2000	MVA-BN	LC16
Immunocompromised	✓	—	—
History of atopic dermatitis	✓	✓	✓
Pregnancy	✓	—	—
Breastfeeding	✓	—	—
Age > 1year	✓	—	—
Allergy to one of the vaccine components	✓	✓	✓
Underlying Heart disease (e.g., coronary artery disease, cardiomyopathy)	✓	—	—
Cardiac risk factors (e.g., hypertension, diabetes, smoking)	✓	—	—

**Table 4 vaccines-10-01419-t004:** Reported adverse events to the two licensed vaccines for monkeypox.

Adverse Events	Vaccine	
	ACAM2000	JYNNEOS
Blood and lymphatic system disorder		
lymphadenopathy	Y	_
lymph node pain	Y	_
Nervous system disorder		
headache	Y	Y
Respiratory disorders		
dyspnoea	y	_
GIT disorders		
Nausea	Y *	Y
vomiting	Y	_
diarrhoea	Y *	_
constipation	Y	_
General disorders and administration site conditions		
Inadvertent inoculation	Y	_
the presence of a “take” following vaccination	Y	_
injection site pain	Y	Y
injection site purities	Y	Y
fatigue	Y	Y

* ACAM2000-treated subjects included nausea and diarrhoea (14%) as commonly reported GI disorders. GIT: gastrointestinal tract; Y: yes.

## Data Availability

Not applicable.

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
