# Peer review of "Preventing the Next Pandemic: Is Live Vaccine Efficacious against Monkeypox, or Is There a Need for Killed Virus and mRNA Vaccines?"

_vaccines, 2022, doi:10.3390/vaccines10091419_

Round 1
Reviewer 1 Report
This manuscript is nicely written. The authors address an important medical condition, which appears to be another threat to human beings on the horizon already. I would like to draw attention to the following points: 1) Prevention and prophylaxis section, second paeragraph: "The COVID-19 pandemic was not controlled until effective vaccines were developed...."Actually, the COVID-19 pandemic is not yet under control. Of course, vaccination has dramatically decreased the number of deaths and hospitalizations, but a probability of acquiring the disease still exists even in the 4th dose-boostered patients (Bar-Yon et al. N Engl J Med 2022; DOI: 10.1056/NEJMoa2201570). Perhaps, the authors might rewrite this phrase; 2) is there a need for another vaccine? section. The authors concluded that "there is still a need to develop an efective and safe new generation of vaccines specific for monkeypox that are killed or mRNA....". However, monkeypox disease has a mortality of 3-6%, as the authors concede. I think that this point deserves further consideration. Could we have time to wait for a mRNA vaccine for monkeypox disease? Would not be better to provide MVA-BN vaccination taking into account its safety profile ( mainly the lack of vaccine-induced myocarditis), and its efficacy (similar to other smallpox vaccines) against a disease with a high mortality rate, which appears to be 95% transmitted by sexual contact (Thornhill et al. N Engl J med 2022; DOI: 10.1056/NEJMoa2207323) ?
Author Response
Reviewer 1’s Comments
Comment 1: Prevention and prophylaxis section, second paragraph: "The COVID-19 pandemic was not controlled until effective vaccines were developed...."Actually, the COVID-19 pandemic is not yet under control. Of course, vaccination has dramatically decreased the number of deaths and hospitalizations, but a probability of acquiring the disease still exists even in the 4th dose-boostered patients (Bar-Yon et al. N Engl J Med 2022; DOI: 10.1056/NEJMoa2201570). Perhaps, the authors might rewrite this phrase.
Response: Thank you for highlighting this comment. We totally agree with your suggestion. We edited this paragraph as follows: “The burden of COVID-19 pandemic was not reduced, in terms of associated morbidity and mortality, until effective vaccines were developed and disseminated.”
Comment 2: Is there a need for another vaccine? section. The authors concluded that "there is still a need to develop an effective and safe new generation of vaccines specific for monkeypox that are killed or mRNA....". However, monkeypox disease has a mortality of 3-6%, as the authors concede. I think that this point deserves further consideration. Could we have time to wait for a mRNA vaccine for monkeypox disease? Would not be better to provide MVA-BN vaccination taking into account its safety profile (mainly the lack of vaccine-induced myocarditis), and its efficacy (similar to other smallpox vaccines) against a disease with a high mortality rate, which appears to be 95% transmitted by sexual contact (Thornhill et al. N Engl J med 2022; DOI: 10.1056/NEJMoa2207323).
Response: We would like to thank you for raising this great point. We might haven’t been able to clarify this point in the previous version of the manuscript. However, what we meant, in this section, was highlighting the need for a new MPX-specific vaccine, but until this happens, other effective and safer vaccines can be used (namely MVA-BN). We added a paragraph at the end of this section as follows: “Nonetheless, until such vaccines are tested and made available, the MVA-BN vaccine can be used to limit the spread of the disease because of its efficacious and safer profile compared to the ACAM2000 vaccine.”
Reviewer 2 Report
Thanks for an interesting comparision between the three possible monkey pox vaccines available in the US.
I have a number of comments:
Page 3: Using the wording "era of pandemics", especially only based on one old reference is misleading, as only a limited amount of pandemics have occured.
Page 4: Stating that monkey pox has the same severity and mortality than smallpox with only a reference from the 1970 is untrustworthy.
Page 4: The statement "Current vaccines cannot protect against MPV as it has a broad host range" does not make sense.
Page 5: The risk of possible unknown long-term averse effects of mRNA vaccines should be mentioned. Moreover, the statements of lifelong immunity is untrustworthy.
Page 5: Stating necessity of booster for ACAM2000 is in contradiction with stating earlier that live vaccines do not need boosters.
Author Response
Reviewer 2’s Comments
Comment 1: Page 3: Using the wording "era of pandemics", especially only based on one old reference is misleading, as only a limited amount of pandemics have occurred.
Response: Thank you for highlighting this point. We agree that this was not the best choice of words. We edited it to “in this era of newly emerging and re-emerging diseases”. We also added a new supporting citation [Peters MA, 2022, Educational Philosophy and Theory].
Comment 2: Page 4: Stating that monkey pox has the same severity and mortality than smallpox with only a reference from the 1970 is untrustworthy.
Response: Thank you again for pointing out this very important point. We have reviewed the literature carefully and based on available evidence, the recent monkeypox outbreak is associated with 3-6% mortality rate (as of June 2022), which is lower than the reported smallpox-associated mortality rate. Therefore, we edited this sentence as follows: “MPX has similar presentation and severity to smallpox but it is associated with lower mortality (3-6% in the recent multi-country outbreak) and less human-to-human transmission.”
Comment 3: Page 4: The statement "Current vaccines cannot protect against MPV as it has a broad host range" does not make sense.
Response: Thank you for your informative comments and for helping improve the quality of the manuscript. We agree with your suggestion; therefore, we deleted this statement and replaced it with the following one: “Current vaccines, although provide cross-protection against monkeypox, are not specific for the causative virus and their efficacy, in the light of the recent multi-country outbreak, is still to be confirmed.”
Comment 4: Page 5: The risk of possible unknown long-term adverse effects of mRNA vaccines should be mentioned. Moreover, the statements of lifelong immunity is untrustworthy.
Response: Thank you for your comments. As for the statement “lifelong immunity” we changed it to “producing durable immune memory”. We added a sentence regarding the risk of long-term AEs of mRNA vaccines as follows: “Noteworthy, a concern associated with this type of vaccine is the unknown risk for long-term adverse events [20].”
Comment 5: Page 5: Stating necessity of booster for ACAM2000 is in contradiction with stating earlier that live vaccines do not need boosters.
Response: Thank you for your comment. We believe that we did not state “live vaccines do not need booster doses” in our review. If you think otherwise, can you please confirm that with a quote from text? That would be greatly appreciated.
That being said, based on the data provided by the US-HHS, inactivated vaccines provide lower immunity compared to live vaccines, and therefore, there might be a need in a booster dose. And, we made this part clear in the revised text as follows: “However, they are insufficient as a single dose and they usually do not provide immunity that is as strong as live vaccines; therefore, booster doses may be needed periodically to produce a sufficient immune response [17].”
Reviewer 3 Report
This review describes the usefulness and disadvantages of current effective vaccines against monkeypox virus infection.
This review is useful and interesting but its organization can be improved because some information is not well placed to be easily understood by a reader who does not know the subject.
In particular, the different generations of vaccine should be presented before naming the vaccine, whereas the manuscript talks about vaccine generations in chapter 4 and defines them in chapter 5.
Afterwards, some sentences are not well understood or need improvement or clarification:
Chapter 2:
“innate immunity is the first-line defence line against infections” : is it correct ?
Chapter 4:
1) “Booster doses are given every three years for people exposed to high virulent strains and ten
years for those exposed to low virulent strains” and
“A booster dose is required every two years for people with high virulent strains and every ten years for those in contact with low virulent strains” : precise which viruses
2) “LC16 is produced from the Lister strain containing a deleted mutation in the viral protein” : which protein ?
Chapter 5:
“cross-rection with the MPV” : cross-reaction
« Not all of these vaccines are not primarily used” : is it correct ?
“Evidence indicates that remarkable advances have been introduced to vaccine technology to enhance after smallpox to enhance the safety and efficacy of these vaccines.” : is tit correct ?
“However, no evidence was found regarding its use for MPV prevention” and
“It should be noted that these vaccines are usually efficacious in preventing MPV infection when used as pre-exposure approaches” : the 2 sentences are contradictory; is it human studies for all ?
Chapter 6 :
“Unvaccinated individuals who … have close contact with a vaccine recipient are at an increased risk for serious problems if they become infected with the vaccine virus either by being vaccinated or by being in close contact with a vaccinated person” : this sentence could be improved
Lastly, some reference are incomplete (25) or do not contain the information indicated where they are placed (21)
Author Response
Cover Letter
Editor-in-Chief
Vaccines Journal-MDPI
August 18th, 2022
Dear Editor,
Thank you for your letter and the opportunity to re-revise our paper on ‘Preventing the Next Pandemic: Is Live Vaccine Efficacious Against Monkeypox, or There is a Need for Killed Virus and mRNA Vaccines?’ The suggestions offered by reviewer 3 have been immensely helpful in improving the quality of our manuscript.
I have included the reviewer’s comments immediately after this letter and responded to them individually, indicating exactly how we addressed each concern or problem and describing the changes we have made. The revisions have been approved by all authors and I have again been chosen as the corresponding author. All edits that have been applied to the revised version of the manuscript have been highlighted in yellow and they can also be tracked through the “Track Changes” option on Microsoft word.
We have revised manuscript as per your comments and ready to make more changes if you suggest. We thank you for your continued interest in our research.
Sincerely,
Ranjit Sah
Tribhuvan University Teaching Hospital
Institute of Medicine, Kathmandu, Nepal
ranjitsah@iom.edu.np
Corresponding author
Reviewer 3 Comments:
Comment 1: This review is useful and interesting but its organization can be improved because some information is not well placed to be easily understood by a reader who does not know the subject. In particular, the different generations of vaccine should be presented before naming the vaccine, whereas the manuscript talks about vaccine generations in chapter 4 and defines them in chapter 5.
Response: Thank you for your comments and for your help in improving the quality of our manuscript. We agree with your suggestion. Therefore, we moved the content of chapter 4 into chapter 5 and vise versa. Thank you!
Chapter 2
Comment 2: “innate immunity is the first-line defence line against infections” : is it correct ?
Response: Thank you for highlighting this mistake. We corrected it as follows: “the first defense line against infection”
Chapter 4
Comment 3: “Booster doses are given every three years for people exposed to high virulent strains and ten years for those exposed to low virulent strains” and “A booster dose is required every two years for people with high virulent strains and every ten years for those in contact with low virulent strains”: precise which viruses
Response: Thank you for your comment. We edited this part as follows: “Booster doses are given every three years for people exposed to high virulent strains of orthopoxviruses and ten years for those exposed to low virulent strains (i.e., Vaccinia virus or Cowpox virus) [21].” And “A booster dose is required every two years for people with high virulent orthopoxviruses and every ten years for those in contact with low virulent strains [25].”
Comment 4: “LC16 is produced from the Lister strain containing a deleted mutation in the viral protein” : which protein ?
Response: Thank you for your comment. This part has been updated as follows: “LC16 is produced from the Lister strain containing a deleted mutation in its immunogenic membrane protein B5R.”
Chapter 5
Comment 5: “cross-rection with the MPV”: cross-reaction
Response: Thank you for noticing this typo. It’s been corrected.
Comment 6: « Not all of these vaccines are not primarily used”: is it correct?
Response: Thank you for noticing this mistake. It has been corrected as follows: “not all of these vaccines are primarily used..”
Comment 7: “Evidence indicates that remarkable advances have been introduced to vaccine technology to enhance after smallpox to enhance the safety and efficacy of these vaccines.”: is it correct?
Response: Thank you for highlighting this mistake. It has been corrected as follows: “Evidence indicates that remarkable advances have been introduced to vaccine technology after smallpox to enhance the safety and efficacy of these vaccines.”
Comment 8: “However, no evidence was found regarding its use for MPV prevention” and “It should be noted that these vaccines are usually efficacious in preventing MPV infection when used as pre-exposure approaches” : the 2 sentences are contradictory; is it human studies for all ?
Response: Thank you for highlighting this point. We removed the first statement and replaced it with this one: “Although LC16 is the only approved smallpox vaccine for use in children, there are no data on its efficacy in preventing MPV infection.”
Chapter 6
Comment 9: “Unvaccinated individuals who … have close contact with a vaccine recipient are at an increased risk for serious problems if they become infected with the vaccine virus either by being vaccinated or by being in close contact with a vaccinated person” : this sentence could be improved
Response: Thank you for your comment. We admit that the statement was redundant. We have edited it as follows: “Unvaccinated individuals (i.e., pregnant women, people with heart or immune diseases, or individuals with skin problems like eczema, dermatitis, or psoriasis) are at an increased risk of serious problems if they become infected with MPV either through vaccination or through close contact with a vaccine recipient.”
Comment 10: Lastly, some references are incomplete (25) or do not contain the information indicated where they are placed (21)
Response: Thank you for your comment. Reference number 21 has been replaced with reference number 52 [references are updated since the content of certain chapters was moved to another chapter]. As for reference 25, this citation was only published on ResearchGate and it has not been peer-reviewed nor published in a PubMed-indexed journal.
